# Relationship between External and Internal Workloads in Elite Soccer Players: Comparison between Rate of Perceived Exertion and Training Load

**Alessio Rossi [1,*], Enrico Perri [2], Luca Pappalardo [3], Paolo Cintia [1] and Fedon Marcello Iaia [2]**

[1] Department of Computer Science, University of Pisa, 26127 Pisa, Italy; paolo.cintia@gmail.com

[2] Department of Biomedical Science for Health, University of Milan, 20122 Milan, Italy; enrico.perri@unimi.it (E.P.); marcello.iaia@unimi.it (F.M.I.)

[3] Istituto di Scienza e Tecnologie dell'Informazione "A. Faedo" of the National Research Council, 56127 Pisa, Italy; luca.pappalardo@isti.cnr.it

* Correspondence: alessio.rossi2@gmail.com

**Abstract:** The use of machine learning (ML) in soccer allows for the management of a large amount of data deriving from the monitoring of sessions and matches. Although the rate of perceived exertion (RPE), training load (S-RPE), and global position system (GPS) are standard methodologies used in team sports to assess the internal and external workload; how the external workload affects RPE and S-RPE remains still unclear. This study explores the relationship between both RPE and S-RPE and the training workload through ML. Data were recorded from 22 elite soccer players, in 160 training sessions and 35 matches during the 2015/2016 season, by using GPS tracking technology. A feature selection process was applied to understand which workload features influence RPE and S-RPE the most. Our results show that the training workloads performed in the previous week have a strong effect on perceived exertion and training load. On the other hand, the analysis of our predictions shows higher accuracy for medium RPE and S-RPE values compared with the extremes. These results provide further evidence of the usefulness of ML as a support to athletic trainers and coaches in understanding the relationship between training load and individual-response in team sports.

**Keywords:** sports analytics; external workload; training volume; internal workload; sports data science

## 1. Introduction

Monitoring training load is a fundamental process to maximize the physical capacity of athletes and to manage their fatigue throughout the season [1]. An athlete's training load can be quantified by external (e.g., global position system and video analysis) and internal parameters (e.g., rate of perceived exertion, heart rate, and lactate). The external training load represents the dose performed, while the internal training load reflects the psycho-physiological response of the athlete [2–5].

The game-based nature of team sports can generate inter-individual variation in external training load, resulting in different internal training loads [1,6]. In soccer the number of matches played during the season together with the inter-individual variation related to physical levels, role positions [7,8], and technical and tactical requirements can lead to a training imbalance, leaving some athletes at risk of overtraining and others failing to reach an adequate training stimulus that could potentially enhance the risk of injuries [9,10]. Therefore, the implementation of monitoring models

able to understand which specific training doses should be applied to individual athletes, and which markers of external load influence the athletes' internal load should be studied [11,12]. Indeed, several studies show that the relationship between planned and perceived training load is weak: the training sessions designed to be easy or intermediate are perceived as harder by players, while the sessions designed to be hard are perceived as less intense [13,14].

Recently, the relationship between internal- and external-load parameters has been investigated to demonstrate that both rate of perceived exertion (RPE) and training load (S-RPE) are significantly related to several indicators of external physical load (i.e., high-speed running distance (>14.4 km/h), number of impacts, and number of accelerations (>3 m·s$^{-2}$) [12]. Furthermore, the magnitude of the within-individual correlations significantly reduces when adjusted for the effects of the other variables, and this may reflect the fact that many additional factors may contribute to the perception of intensity in intermittent team-sport exercise [6]. Similarly, other studies provide new evidence to support the use of global position system (GPS) data as a valid global indicator of training responses and intensity in elite rugby, Australian football, and in sub-elite soccer [6,15,16]. For this reason, the use of new advanced sensing systems that allow predicting RPE and S-RPE based on external load could be a useful tool to improve training periodization. Previous studies demonstrated that measures which derivate from RPE and S-RPE could be useful tools for monitoring training loads. Hence, assessing external workloads becomes crucial to optimize athlete-monitoring practices [17,18].

The main aim of this study is to provide a machine learning (ML) model that is able to predict the players' internal load by external ones extracted from GPS raw data. This ML model also allows athletic trainers and coaches to detect the discrepancy between the effort that the player should have perceived for a specific external load and the effort that the player has perceived, hence highlighting possible problems induced by training periodization. Based on the cited papers, we hypothesized that it is possible to predict internal load by the external one accurately and that we can further improve the prediction by taking into consideration contextual features such as physiological aspects and individual characteristics. Moreover, the main goal of this paper is to improve the understanding of the discrepancy between the internal load recorded and the one predicted, so to help athletic trainers and coaches better periodize training, reducing the stress of the players while maximizing the effect of training.

## 2. Materials and Methods

### 2.1. Data Collection and Feature Extraction

Twenty-two elite soccer players (age = 21.96 ± 4.53 years; height = 180.68 ± 5.23 cm; weight = 72.36 ± 4.19 kg) competing in an Italian championship during the 2016/2017 season were recruited in this study. All movement players were included in this study: central backs (*n* = 3), fullbacks (*n* = 5), midfielders (*n* = 5), wingers (*n* = 5), and forwards (*n* = 4). Goalkeepers were excluded from this study. Participants gave their written informed consent to participate in the study.

Players' physical activity was recorded during 160 training sessions and 35 matches by using a portable non-differential 10 Hz global position system (GPS) (Playertek, Dundalk, Ireland) that is also characterized by a 400 Hz Tri-Axial Accelerometer and 10 Hz Tri-Axial magnetometers [19]. The GPS devices were placed between the players' scapulae through a tight vest and were activated 15 min before the data collection, in accordance with the instructions of the manufacturer to optimize the acquisition of satellite signals. A total of 1674 individual sessions corresponding to 195 collective sessions performed during the 2016/2017 season was recorded. Eighty-eight (88) workload indicators—21 kinematic (e.g., distance in m or time covered during the training session at specific velocity), 37 metabolic (e.g., distance in m or time covered with a metabolic power above specific thresholds expresses as watt per kg) and 30 mechanicals (e.g., number or m of accelerations/decelerations above specific thresholds)—were extracted from the GPS data. Moreover, the players' rate of perceived exertion (RPE) was collected in isolation about 30 min after the training sessions and matched using the CR-10 Borg scale [20]. This scale is a simple numerical list where 0

refers to "No exertion at all", while 10 refers to "Maximal exertion". The players were asked to tell their rate of exertion (i.e., overall physical stress and fatigue) during the activity. Moreover, the RPE is multiplied to the duration of the training session to obtain the training load (S-RPE) of each training session per player. In this study, this value is divided in a set of ranked data of ten equally large subsections.

In accordance with the Helsinki Declaration as revised in 2013, the study was approved by the Ethical Committee of the University of Milan that do not release any document because it didn't consider the formal approval necessary for this study.

## 2.2. Feature Engineering

In addition to the anthropometric players' characteristics (e.g., age, body mass index, and role) and the 88 daily features extracted directly from the GPS data, 182 individual features and 11 contextual features were constructed. Table 1 describes the features used in our study.

The individual features reflect the personal characteristics of the players and their training workloads. Note that these features include both the players' workload in the current training session and a summary of his workloads in the previous sessions. All the individual features were normalized using the z-score to reduce intra-subject variability.

First, to take into account a player's effort during the previous weeks, the exponential weighted moving average (EWMA) with a span of 7 days for each of the 88 features extracted from GPS data was computed. The EWMA allows for weighting more workloads performed close to the current training session than workloads performed in days long before the current training session. Second, for each of the 88 daily features extracted from GPS data, the acute:chronic workload ratio (ACWR) and the acute workload were computed [21]. A player's ACWR is defined in the literature as the ratio between his acute workload and his chronic workload. As proposed by Murray et al. [21], the acute workload of a player was computed as EWMA of his workload in the previous 7 days; whereas, the chronic workload of a player was computed as the EWMA of his workload in the previous 28 days [21].

Third, for each of the 88 features extracted from the GPS data, the monotony measure was computed. Monotony is defined as the training variation (i.e., the ratio between the mean and standard deviation of a time interval) across 6 and 28 days, while the strain measure is defined as the training stress across 6 and 28 days [22]. Finally, to take into consideration the perceived exertion of previous days, the rolling mean of the RPE observed in the previous 6 days was computed.

In addition to the individual features, some contextual features such as the results of the previous match, the mean of the RPE perceived by the team in the previous soccer activity, and the mean of RPE by role were considered. The training day where players perform the session was added to the list of features, so as to take into account the fact that players perform different workloads in accordance with the distance from the previous and next matches. Finally, since the fatigue could be perceived differently as the season goes by, the month where players perform the training session was also added.

**Table 1.** Description of the individual and contextual features extracted from global position system (GPS) data and the players' personal features collected during the study.

| Individual Features | |
|---|---|
| Personal | Age, height, weight, body mass index (BMI), and role |
| Daily | 88 GPS features of a training/match extracted from the GPS device (i.e., 21 kinematic, 37 metabolic, and 30 mechanical) |
| ACWR | Acute:chronic workload ratio − ACWR − (i.e., exponential weighted moving average (EWMA) with span = 6/EWMA with span = 28) of the 88 daily features (i.e., 21 kinematic, 37 metabolic, and 30 mechanical) |
| Acute | Acute workload (EWMA with span = 6) of the 88 daily features (i.e., 21 kinematic, 37 metabolic, and 30 mechanical) |
| Monotony$_{Week}$ | Reflection of training variation across a week (6 days). It is the ratio between training loads (i.e., product between duration and rate of perceived exertion (RPE)) mean performed in one week and its standard deviation. |
| Monotony$_{Month}$ | Reflection of training variation across a month (28 days). It is the ratio between training loads (i.e., product between duration and RPE) mean performed in one month and its standard deviation. |
| Strain$_{Week}$ | Reflection of overall training stress from the week. It is the product between the training loads (i.e., product between duration and RPE) mean and the Monotony$_{Week}$. |
| Strain$_{Month}$ | Reflection of overall training stress from the week. It is the product between the training loads (i.e., product between duration and RPE) mean and the Monotony$_{Month}$. |
| RPE$_{PrevPlayer}$ | Mean of the RPE provided by a player in the previous week (6 days) |
| Contextual Features | |
| Win-Draw-Loss | Results of previous match |
| RPE$_{PrevTeam}$ | Mean of the RPE provided by the team in the previous day |
| RPE$_{PrevRole}$ | Mean of the RPE provided by the players with the same role in the previous day |
| ID training | Seven Boolean features reflect the day of the week when players perform a training (i.e., md = match day, md + 1 = day after a match, md + 2 = two days after a match, md − 4 = four days before a match, md − 3 = three days before a match, md − 2 = two days before a match, and md − 1 = one day before a match) |
| Month | Month when a training was performed |

*2.3. Prediction of RPE and S-RPE through ML*

In this study, an ML classifier was constructed by using a training dataset where each example refers to a single player's training session, and consists of a vector of the workload features that describe the player's recent workload, including the current training session and both the RPE and S-RPE labels. Our classifier is based on ordinal regression (i.e., a regression analysis used for predicting an ordinal variable) [23].

### 2.3.1. Construction of Training Dataset

Given a feature set S, the training dataset $T_S^{RPE}$ and $T_S^{S\text{-}RPE}$ for the learning task is constructed by a two-step procedure:

1.  For every individual training session i, a feature vector $\boldsymbol{m_i} = (h_1; \ldots; h_k)$ where $h_j \in S$, $(j = 1; \ldots; k)$, is a training workload feature and $k = |S|$ is the number of features considered, was constructed. All the feature vectors compose matrix $F_S = (\boldsymbol{m_1}; \ldots; \boldsymbol{m_n})$, where $n$ is the number of individual training sessions and matches in our dataset ($n = 1674$);

2.  In $T_S^{RPE}$, every feature vector $\boldsymbol{m_i}$ is associated with the RPE provided by each player at the end of the training sessions and matches, while in $T_S^{S\text{-}RPE}$ each vector is associated with a S-RPE class (i.e., product between RPE and duration of the session; training load) that is defined as the decile in which the S-RPE values are grouped taking into consideration all the training loads recorded during the entire season. Hence, matrix $F_S$ is associated with a vector of labels $c = (c_1, \ldots, c_n)$ (one for each training session). The training dataset for the learning task is finally $T_S = (F_S, c)$.

### 2.3.2. Experiments

First of all, for both $T_S^{RPE}$ and $T_S^{S\text{-}RPE}$, a feature selection process was performed on 20% of the dataset randomly selected to determine the most relevant features for classification. This process was performed to reduce the dimensionality of the feature space and the risk of overfitting, allowing a more straightforward interpretation of the machine learning models, due to the lower number of features [24]. To this aim, a recursive feature elimination with cross-validation (RFECV) [25] was used to select the best subset of features. RFECV is a wrapper method for feature selection [26], which initially starts by training a predictive model (a decision tree in our experiments) with all the features in the feature set. Then at every step, RFECV eliminates one feature, trains the decision tree on the reduced feature set, and calculates the score on the validation data. For each feature, its weight was extracted from the constructed ordinal regressor. The weights range in the interval [−∞, +∞], where values lower than 0 negatively affect the RPE or S-RPE, while vice-versa for the positive values. The subset of features producing the maximum score on the validation data is considered to be the best feature subset [25]. In addition to the ordinal regressor (ordinal), the following state-of-the-art classifiers were also constructed: decision tree regression (DT), random forest regression (RF), epsilon-support vector regression (SVR), logistic regression (logit), K-nearest neighbors (KNN), and linear regression (LR). Even if several other ML approaches were tested during analysis, we provide the results of the best ones.

Our classifiers were validated on the remaining 80% of the dataset with a 3-fold stratified cross-validation strategy [27], stratified by player identification (ID); the dataset was divided into 3 folds. For each fold, 90% of the dataset was used as a training set and 10% of it as a test set. Each fold was made by preserving the percentage of samples for each class. Thus, each sample in the dataset was tested once, using a model that was not fitted with that sample. The goodness of the classifiers was measured by a root mean squared error (RMSE) and mean of absolute difference (MAD). Low values of RMSE and MAD indicate a high accuracy of the model. The goodness of our models was also assessed by the relationship (i.e., Pearson correlation coefficient) between RPE or S-RPE observed and predicted. The Pearson correlation coefficient (r) can take a range of values from −1 (negative correlation) to +1 (positive correlation). In addition, the agreement between the real effort perceived and the predicted one was assessed using Bland–Altman analysis [28]. This analysis was used in order to assess the bias (i.e., mean difference between RPE or S-RPE observed and predicted) and the systematic error (i.e., the relationship among the mean and the difference between RPE or S-RPE predicted and observed) of our classifiers compared to the perceived exertion filled by the players.

Finally, our predictive models were compared with two baselines: baseline $B_1$ randomly assigns a class to an example by respecting the distribution of classes; baseline $B_2$ always assigns to an example the majority class.

## 3. Results

Descriptive statistics of RPE, S-RPE, and the GPS features are provided in supplementary S1 Appendix, Figure S1 and Figure S2. For $T_s^{RPE}$, the feature selection process selected 53 features out of 286, whereas for $T_s^{S-RPE}$, it selected only 11 features. Figures S3 and S4 show the importance of the features in the classifiers computed according to a coefficient measuring how much each variable contributes to explain the RPE and the S-RPE, respectively [12].

Although just a small subset of features is selected, in $T_s^{RPE+RFECV}$ we observe that the information derived directly from the training (i.e., daily workloads features), from the last six days of training (i.e., acute workloads features) and from the discrepancy between acute and chronic workloads (i.e., ACWR workload features) is related to the perceived exertion. The acute features affect the RPE detection (1.22 ± 0.49 absolute weights mean) more than daily and ACWR features (1.19 ± 0.59 and 1.10 ± 0.58 absolute weights mean, respectively). These results show that the workloads performed in the previous week are predictive of the effort during the training or match. Moreover, even if the influence of ACWR features are lowest compared to acute and daily ones, these features contribute to increasing the ability of the ML model to predict the internal load, but, as suggested by previous studies, these features could not be used to predict injuries [10,29]. Furthermore, metabolic features (e.g., time in power zone between 0–5 W/kg and 5–10 W/kg) affect the RPE detection (1.27 ± 0.65 absolute weights mean) more than kinematic (e.g., duration of the training session) and mechanical ones (e.g., time in acceleration zone 0–1 m/s²) (1.18 ± 0.43 and 1.10 ± 0.52 absolute weights mean, respectively). See Figure S3 for a detailed list and the importance of all the features used in the ML approach. Regarding the ID training feature, in $T_s^{S-RPE+RFECV}$ it is possible to observe that the information derived directly from the training (i.e., daily workloads features) is the only one related to S-RPE. These results show that the training load (S-RPE) is only affected by the workload performed in that specific session. Moreover, the mechanical features (e.g., distance in acceleration zone 1–2 m/s²) affect the S-RPE detection (0.76 ± 0.34 absolute weights mean) more than kinematic (e.g., distance per minute during the training session) and metabolic ones (e.g., time in power zone between 0–5 W/kg) (0.59 ± 0.01 and 0.30 ± 0.09 absolute weights mean, respectively).

Figure S5 shows the performance of the classifiers both for $T_s^{RPE+RFECV}$ and $T_s^{S-RPE+RFECV}$. The ordinal regressor is the best classifier in describing both the players' RPE and S-RPE. Indeed, the ordinal regressor has lower values of RMSE and MAD compared to the two baselines. A moderate-high correlation (r = 0.70) between the RPEs observed, and RPEs predicted by the ordinal regressor was found. Figure 1a shows that, 72.5% of times, the ordinal regressor reported high RPE values when players perceived less than four of RPE, and 87% of times it reported low RPE values when players perceived more than seven RPE. Similarly, a high correlation (r = 0.84) was detected between the observed S-RPEs and the predicted ones showing that 50.8% of times the ordinal regressor overestimated (i.e., the algorithm provided an S-RPE higher than the observed one) the S-RPE classes lower than four and 49.5% of times it underestimated (i.e., the algorithm provided an S-RPE lower than the observed one) the S-RPE classes higher than seven (Figure 1b). These results are corroborated by the Bland–Altman analysis provided in Figure 2, in which we detect a low bias between the observed and the predicted RPE (0.01 ± 1.20 arbitrary units (AU); see Figure 2a) and between the observed and the predicted S-RPE (0.002 ± 1.57 AU; see Figure 2b). Moreover, the Bland–Altman analysis also shows a lower systematic error in S-RPE (r = −0.13; see Figure 2b) compared to RPE (r = −0.30; see Figure 2a). The negative value of systematic error suggests that the predicted RPE and S-RPE are higher than the observed ones when players perceived low RPE while it is lower when the players perceived a high RPE.

Finally, Figure 3 shows that our model understates the RPE and S-RPE class when they show a value higher than seven in days close to the match, while it overestimates the players' effort in the day before a match when players perceived less than four. A similar distribution of both underestimations and overestimations was recorded when the players' perceived effort was between four and seven.

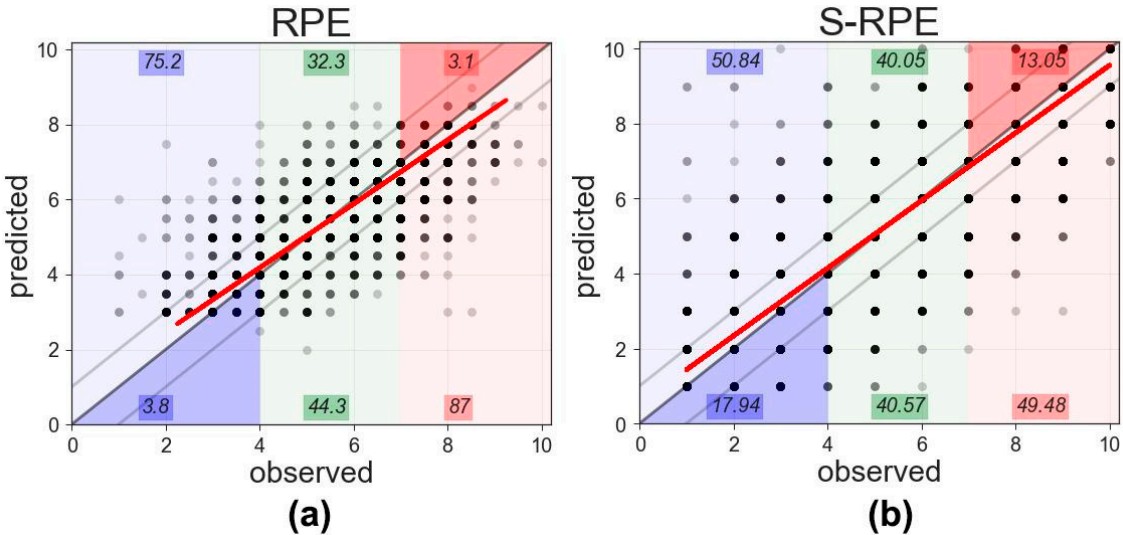

**Figure 1.** Correlation analysis. (**a**) Relationship between the observed and the predicted rate of perceived exertion (RPE). (**b**) Relationship between the observed and the predicted training load (S-RPE). The blue area refers to observed values lower than 4, the green area refers to observed values between 4 and 7, and the red area refers to observed values higher than 7. The values provided in the boxes reflect the percentage of overestimations and underestimations above 1 standard deviation (grey line) of the ordinal regressor in relation with the observed values. The red line refers to the trend line of the relationship between the values observed and predicted.

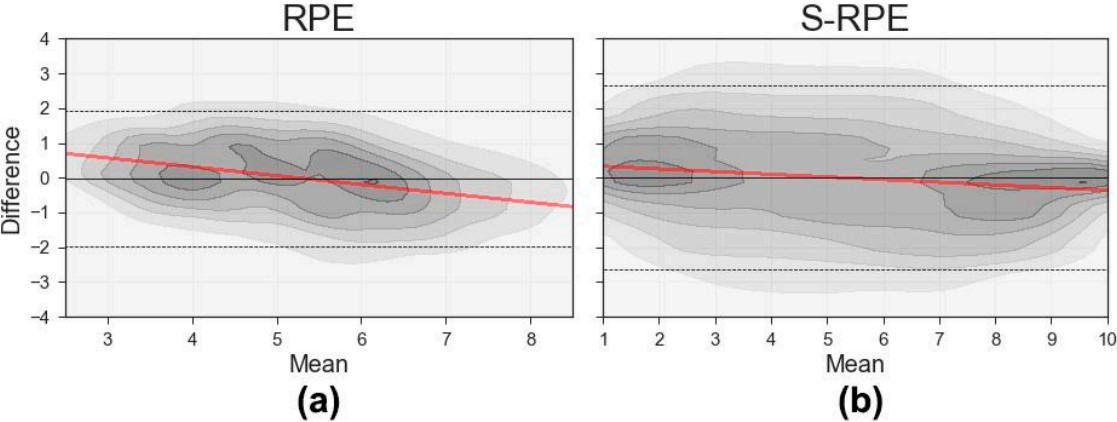

**Figure 2.** Bland–Altman analysis. Relationship between the mean and the difference among the observed values and the ones predicted by the ordinal regressor. (**a**) and (**b**) refer to RPE and S-RPE, respectively. The black line refers to the mean of the difference (i.e., bias), the black dot line reflects the 1.96 standard deviation to the mean of the difference (i.e., confidence interval), and the red line reflects the relationship between mean and difference (i.e., systematic error).

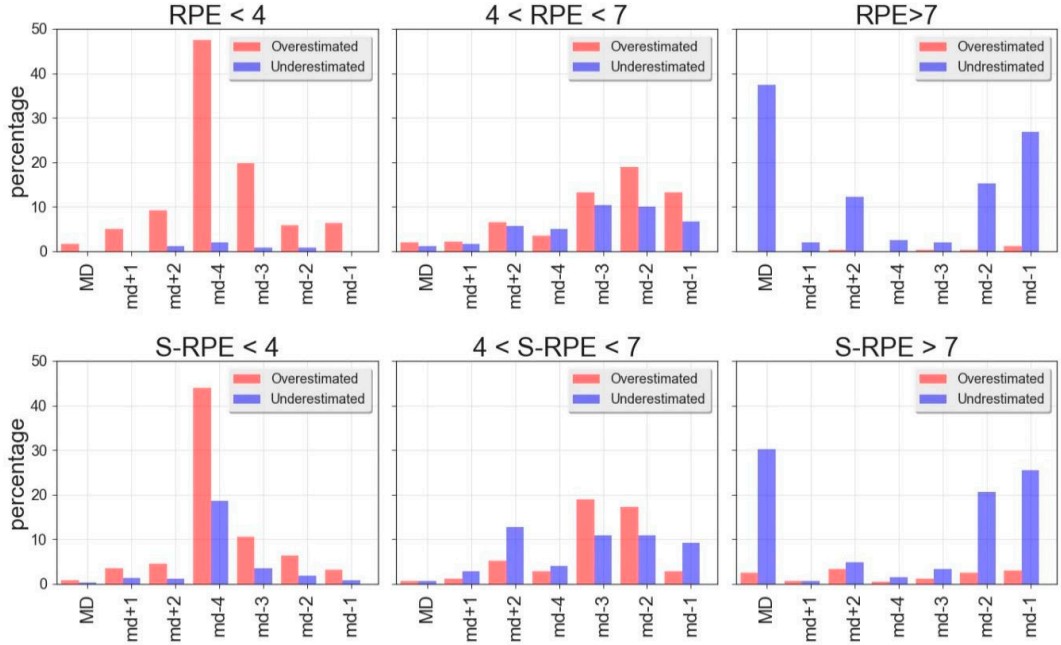

**Figure 3.** Distribution of discrepancy during soccer training week. The distribution of under-estimation (i.e., the algorithm provided a value lower than the observed one) and over-estimation (i.e., the algorithm provided a value higher than the observed one) in three different RPE and S-RPE class ranges (i.e., <4, between 4 and 7, and >7). MD refers to the distance to previous (+) and following (−) match day.

## 4. Discussion

RPE is a measure of internal load widely used in team sports because it is easy to record and assess, it has a low cost, and reflects the external load [17]. In several team sports, it was demonstrated that training load variables (e.g., total distance, high speed running, and accelerations) influence RPE [8,12–19,21–31]. Our study confirms that external loads recorded during the training sessions reflect the perceived exertion of the players and their workload (i.e., RPE and S-RPE, respectively). The main result of this study is that we find a discrepancy between the observed RPE and the RPE predicted by the ordinal regressor, especially for extreme RPE values (i.e., <4 and >7). Indeed, while RPE values close to the average (i.e., 4–7) the ordinal regressor is accurate and overestimates the perceived exertion at low values of RPE and underestimates it at high values. Despite several previous studies which detected a strong relationship between perceived exertion and external loads [6,12–16], the data recorded in this study reported an inconsistent relationship between extreme values of RPE and training workload. Hence, our results demonstrate that not only the external features but also the contextual one (e.g., psychological aspects induced by the distance to the matchday) affect the players' perceived exertion. This aspect is crucial for athletic trainers and coaches to understand the response of their player to external stimuli and to plan the future training session to minimize the effect of training workloads and the stress induced by the match day.

Considering that the extensive use of group exercises and the different physical requirements associated with each position could have an impact on both within- and between-player differences in workload training, the quantification of internal and external workload can be a valid method to monitor training sessions. Another exciting result of this study suggested that RPE and S-RPE are strongly affected by the volume of the total training and weakly affected by the training intensity (as also suggested by Gaudino et al. [12]). Accordingly, a previous study showed that in elite rugby players the RPE is mostly explained by the volume of the training workload (i.e., total distance covered in a session) and by the stress sustained during the physical activity (i.e., a combination of accelerations, decelerations, changes of direction, and impacts) [6]. Moreover, in elite soccer players, the S-RPE is weakly reflected by the high-intensity volume (i.e., high-intensity running) [12–19,21–

31] as well. These cited studies suggest that the session's internal load is reflected by the external load, but the S-RPE provides additional information of the players that the external load could not take into consideration. A previous study showed that it is possible to derive a more robust index than RPE to assess internal load during a training session by multiplying the RPE by the session time [12]. In our study, we highlighted that these two indices reflect different aspects of the training workload. In particular, we show that RPE is related to the stress that the players support in the previous week (see Figure S3), whereas the S-RPE is only affected by the workload performed in the current training session (see Figure S4). Therefore, athletic trainers and coaches could obtain more information about the internal load of their players by assessing both RPE and S-RPE. In this way, they could have a complete overview of the training characteristics that affect internal load to schedule the best training periodization for their athletes [17,18]. The combination of the proposed ML model with an algorithm that automatically generates sport training programs based on internal loads [32,33] could permit to schedule more interpretable training plans based on the assessment of the influence of different parameters on RPE and S-RPE.

This study is the first that investigates the effect of distance to the match (i.e., stress induced by the match) on extreme RPE. In particular, our results show that the highest frequency of overestimation and underestimation is in the days long before the match when players provided RPE and S-RPE <4 (i.e., <474.0 S-RPE), and in the days immediately before the matches and during the matches in RPE and S-RPE >7 (i.e., >726.0 S-RPE), respectively. This result suggests that the exertion perceived by players is affected by the distance in days to the matches. In particular, the RPE and S-RPE increase or decrease in accordance with the match day. One of the most important features in the ML model that affects the players' RPE is the dummy features "MD-4" (i.e., match day minus 4; Figure S3) and one of the best features affecting the players' S-RPE is the dummy features 'MD' (i.e., match day; Figure S4). Hence, the RPE and S-RPE seem to be affected by the psychological tension caused by the official game that has to be taken into consideration by athletic trainers and coaches during the evaluation of the players' internal load during the training sessions.

*Limitations of the Study*

This study has a few limitations. First of all, it is not possible to generalize the results obtained in this study due to the small sample size derived from a single soccer team. Hence, we propose only an approach to evaluate the internal load of the soccer player because of possible differences in physiological responses of an external stimulus, physical characteristics, and training periodization of players competing in different teams. However, in future works, we plan to assess if the results detected in our study are generalizable for all teams, or if effort is perceived in a different way depending on individual and training characteristics. Another limitation of this study is that we do not have disposal data derived from physiological and psychological aspects such as heart rate, blood lactate, sleep quality, anxiety, or stress status that could affect the perceived exertion of an external stimulus. In future works, we plan to increase the number of features recorded in each training session or match to improve the accuracy of our model.

## 5. Conclusions

The ML approach provided in this study permits to automatically and objectively detect the internal load based on external and contextual features. In particular, in this study we highlighted that the RPE and S-RPE are affected by the volume of trainings more than their intensity. As a matter of fact, that RPE is affected by the workload performed in the previous training week, while S-RPE reflects the workload performed in the current training session. Additionally, for external workloads, the psychological tensions (e.g., the distance to the official games) are fundamental aspects affecting both the RPE and S-RPE. The novelty of this study is that it is possible to automatically predict RPE and S-RPE. By using this objective information, athletic trainers and coaches could also evaluate the discrepancy between the exertion perceived by the players and the one that the player should have perceived for a specific training workload. This discrepancy could indicate problems or different

levels of stress of the players that athletic trainers and coaches do take into consideration to accurately monitor the soccer players' workloads and enhance training prescriptions.

**Supplementary Materials:** The following are available online at www.mdpi.com/xxx/s1, S1 Appendix: Descriptive statistics, Figure S1: RPE and S-RPE histogram, Figure S2: Identity card of training workloads, Figure S3: T$_S^{RPE}$ feature selection, Figure S4: T$_S^{S-RPE}$ feature selection, Figure S5. Classifier performances on both $T_S^{RPE+RFECV}$ and $T_S^{S-RPE+RFECV}$.

**Author Contributions:** Conceptualization, A.R., E.P., and F.M.I; methodology, A.R., L.P., and P.C.; validation, A.R.; formal analysis, A.R., L.P., and P.C.; data curation, P.E. and F.M.I.; writing—original draft preparation, A.R., L.P., P.C., E.P., and F.M.I; writing—review and editing, A.R., L.P., P.C., E.P., and F.M.I; visualization, A.R.; supervision, F.M.I.

**Funding:** This work is partially supported by the European Community's H2020 Program under the funding scheme INFRAIA-1-2014-2015: Research Infrastructures grant agreement 654024, www.sobigdata.eu, SoBigData. The funders had no role in study design, data collection and analysis, decision to publish, or preparation of the manuscript. There was no additional external funding received for this study.

**Conflicts of Interest**: The authors declare no conflicts of interest.

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
