# Peer review of "Relationship between External and Internal Workloads in Elite Soccer Players: Comparison between Rate of Perceived Exertion and Training Load"

_applsci, doi:10.3390/app9235174_

Round 1
Reviewer 1 Report
Relationship between external and internal workloads in elite soccer players: comparison between Rate of Perceived Exertion and Training Load
General
I carefully review the manuscript entitled “Relationship between external and internal workloads in elite soccer players: comparison between Rate of Perceived Exertion and Training Load”. Authors established a research question of building a machine learning (ML) model in order to predict the players’ internal load by external workloads extracted from GPS raw data. The findings are encouraging; however, authors have to reconsider some points inside the manuscript before it can be considered for publication.
Was the sample size enough in order to support the construction of a Machine Learning model to predict the players’ internal load by external workloads? How authors can be sure that the results of this study can be generalized to other players? What was the level of the team participated in the study? In line with this, was this relationship the same for all players? Did the authors examine the player’s position to the suggested Machine Learning model?
Authors evaluate and recorded the physical activity of 22 players during 160 training sessions and 35 matches. Do the authors have any evidences about the changes of RPE and sRPE or other values of the Machine Learning model for training and matches separately? In this way coaches and athletes could possibly monitor training and matches separately.
Authors report data for RPE and sRPE, however, are there any data for heart rate variability during training and matches?
Are there any limitations in this study? How authors could be sure that players performed their best during training and competition? How authors can be sure that RPE alone and not with a combination of heart rate or lactate can provide a better Machine Learning model?
As a final point, why are these results important to coaches and players when authors mention inside the discussion that: “The main results of this study suggested that the RPE and S-RPE are strongly affected by the volume of the total training and weakly affected by their intensity (as also suggested by Gaudino et al. [12])”? If these results are also referred by Gaudino et al., then what new can the reader learn by this study? And if this is the case, then I suggest to authors to change the first paragraph of Discussion and present the main finding and discuss the secondary findings of the study in the beginning of the discussion. Authors should clarify the meaning of their Machine Learning model and make it more conceptual to coaches and players.
Line 60: Change out to our.
Line 60 – 61: Is this the hypothesis of the sudy?
Line 64: Are 22 players enough for your statistic analysis? I recommend to authors to add results from a Power analysis test.
Line 78: Why authors evaluate RPE 20 minutes after the training sessions and matches when the recommendations are 30 minutes (Foster et al., 2004; Borresen and Lambert, 2009 & 2010)?
Table Individual features: Acute:Chronic workload ratio (EWMA with span=6/EWMA with span=28) of the 88 daily features (i.e., 21 Kinematic, 37 Metabolic and 30 Mechanical). Please provide some examples for kinematic, metabolic and mechanical features include inside the model.
Line 184: For example, what metabolic features where that?
Line 213: Please delete the “Discussion” at the end.
Lines 214 – 216: Please delete these sentences.
Line 286 - 288: “Hence, these two variables of internal load provides information of both physical and physiological responses permitting to athletic trainers and coaches to accurately monitor the soccer player’s workloads in order to enhance training prescription” Inside the manuscript it is not clear what is the physiological and physical response from the model. Please clarify.
Conclusions: Conclusions should focus on the practical application of the results of this study to coaches and players. Also, authors should conclude their main finding and and tips for sports practice.
Author Response
Point 1. I carefully review the manuscript entitled “Relationship between external and internal workloads in elite soccer players: comparison between Rate of Perceived Exertion and Training Load”. Authors established a research question of building a machine learning (ML) model in order to predict the players’ internal load by external workloads extracted from GPS raw data. The findings are encouraging; however, authors have to reconsider some points inside the manuscript before it can be considered for publication.
Authors’ response: We would like to thank reviewer 1 for the useful comments provided. We will provide detailed answers below. All the changes are highlighted in red throughout the paper.
Point 2.1. Was the sample size enough in order to support the construction of a Machine Learning model to predict the players’ internal load by external workloads? How authors can be sure that the results of this study can be generalized to other players?
Authors’ response: In our study the statistical unit is not the players (n=22) but the number of individual training sessions (n=1674). Hence, the sample size justify the use of a machine learning approach. To avoid any possible influence of the relationship between observations on Ordinal Regression, we also inserted player ID as a feature for the model. However, this information was meaningless, in terms of information gain, for all the models suggesting that the model is generalizable for all the players in the team. Nonetheless, it is not possible to generalize this model to players for other teams because of possible athletes’ difference of physiological response, physical characteristics and training periodization. Future work are required in order to assess if our model is generalizable or it is required a specific model for each team. Due to the fact that this kind of data are extremely difficult to obtain from different teams we are not able to assess if our model it is generalizable. We have added this point on the limitation section inserted at the end of the discussion session and we modified some part of the text to highlight the fact that these results are referred to a specific team and could not be generalized for the population.
Point 2.2. What was the level of the team participated in the study?
Authors’ response: The players recruited for this study competing in an elite Italian championship during season 2016/2017 as asserted in “Data collection and feature extraction” section. We could not give more information about the soccer team because they have expressed the willing of staying anonymous.
Point 2.3. In line with this, was this relationship the same for all players? Did the authors examine the player’s position to the suggested Machine Learning model?
Authors’ response: In addition to external workloads information, we inserted into the model several other features not directly related to external load (e.g., role, player ID, training day and match day) to investigate any possible relation of contextual information. The importance of these features in our models is extremely low, which suggests that the internal load is not affected by this contextual information. For this reason, the development of specific models for this contextual information is not meaningful.
Point 3. Authors evaluate and recorded the physical activity of 22 players during 160 training sessions and 35 matches. Do the authors have any evidence about the changes of RPE and sRPE or other values of the Machine Learning model for training and matches separately? In this way coaches and athletes could possibly monitor training and matches separately.
Authors’ response: As highlighted in the previous answer, due to the fact that the information referring to the match day is useless to improve the internal load prediction, the development of specific models for match and training is useless. Additionally, besides higher RPE and S-RPE values recorded during match day compared to training days, in literature there is not evidence that support the implementation of two different models. Just to double check our findings, we develop ML models for each one of the possible match days (e.g., MD, MD+1 and MD+2) discovering that the model with lower error is the one considering all the MD together (in this later model one of the features used is the information about the match day as explained in the text).
Point 4. Authors report data for RPE and sRPE, however, are there any data for heart rate variability during training and matches?
Authors’ response: Unfortunately, we do not have the availability of heart rate data for training and matches. A future work is going to be scheduled in order to add this information. We have added this point in the text where we highlight the limitations of this study.
Point 5. Are there any limitations in this study? How authors could be sure that players performed their best during training and competition? How authors can be sure that RPE alone and not with a combination of heart rate or lactate can provide a better Machine Learning model?
Authors’ response: We have added a limitations section at the end of discussion one highlighting any possible limitations of our work. One of the huge limitation is the lack of some important information of players’ response such as heart rate and lactate.
Point 6. As a final point, why are these results important to coaches and players when authors mention inside the discussion that: “The main results of this study suggested that the RPE and S-RPE are strongly affected by the volume of the total training and weakly affected by their intensity (as also suggested by Gaudino et al. [12])”? If these results are also referred by Gaudino et al., then what new can the reader learn by this study? And if this is the case, then I suggest to authors to change the first paragraph of Discussion and present the main finding and discuss the secondary findings of the study in the beginning of the discussion. Authors should clarify the meaning of their Machine Learning model and make it more conceptual to coaches and players.
Authors’ response: Thanks for the suggestion. We modify the first part of the introduction section to better highlight the novelty of this study. In particular, in the introduction, we have added the following sentence: “The main aim of this study is to provide a machine learning (ML) model that is able to predict the players’ internal load by external ones extracted from GPS raw data without the need of directly asking to the players.” Moreover, we add the following paragraph to the Conclusion: “The novelty of this study is the possibility to predict RPE and S-RPE automatically without asking directly to the players. Moreover, by using this information, athletic trainers and coaches could also evaluate the discrepancy between the exertion perceived by the players and the one that the player should have perceived for a specific training workload. This discrepancy could indicate problems or different levels of stress of the player that the athletic trainers and coaches could take into consideration to accurately monitor the soccer player’s workloads and enhance training prescription.”
Point 7. Line 60: Change out to our.
Authors’ response: This sentence was deleted during the revision process.
Point 8. Line 60 – 61: Is this the hypothesis of the study?
Authors’ response: This was an anticipation about the results of this study. We modify this part in order to empathize the aim of this study adding some hypothesis of our study.
Point 9. Line 64: Are 22 players enough for your statistical analysis? I recommend to authors to add results from a Power analysis test.
Authors’ response: As explained before, In our study the statistical unit is not the players (n=22) but the number of individual training sessions (n=1674). Hence, the sample size justify the use of a machine learning approach. In our view, the power analysis a-posteriori is useless due to the fact that statistical significance are obtained with this sample size.
Point 10. Line 78: Why authors evaluate RPE 20 minutes after the training sessions and matches when the recommendations are 30 minutes (Foster et al., 2004; Borresen and Lambert, 2009 & 2010)?
Authors’ response: Thanks for this useful comment. This is a typo. We follow the literature recommendations and record the RPE of about 30 minutes after the end of the training sessions and matches. We modify the text in accordance with your suggestion.
Point 11. Table Individual features: Acute:Chronic workload ratio (EWMA with span=6/EWMA with span=28) of the 88 daily features (i.e., 21 Kinematic, 37 Metabolic and 30 Mechanical). Please provide some examples for kinematic, metabolic and mechanical features include inside the model.
Authors’ response: We have added some example in the “Data collection and feature extraction” section for each group of features.
Point 12. Line 184: For example, what metabolic features where that?
Authors’ response: We decided not to show all of this information in the main text in order to make the text fluent for the readers, but we insert only some example of them. If the readers are interesting to study in deep which specific metabolic, mechanic and kinematic features (training characteristic widely know in sport science community – target of this study) are used in the ML model could find all the information in figure S3. We have added a sentence on the text that indicate where the reader could find this information.
Point 13. Line 213: Please delete the “Discussion” at the end.
Authors’ response: Done
Point 14. Lines 214 – 216: Please delete these sentences.
Authors’ response: Done
Point 15. Line 286 - 288: “Hence, these two variables of internal load provides information of both physical and physiological responses permitting to athletic trainers and coaches to accurately monitor the soccer player’s workloads in order to enhance training prescription” Inside the manuscript it is not clear what is the physiological and physical response from the model. Please clarify.
Authors’ response: Thanks for the suggestion. We have modified the text in order to better explain the physiological response from the model.
Point 16. Conclusions: Conclusions should focus on the practical application of the results of this study to coaches and players. Also, authors should conclude their main finding and and tips for sports practice.
Authors’ response: We have modified the “Discussion” and “Conclusion” sections focusing on some practical application for athletic trainers and coaches.
Reviewer 2 Report
The paper has merit, but there are issues that need to be discussed before the paper reaches publication quality level.
Major:
The paper relies on a quite small dataset: although they use 1674 training session, they belong to only 22 players. I wonder whether the analysis (and the drawn conclusions) may suffer from this small number. The experimental design only involves a 3-fold cv + 90/10 stratified split; this design may be not sufficient to warrant reproducibility - following the US-FDA guidelines exploited in the MAQC/SEQC studies, a 10x5-CV + external validation Data Analysis Plan would be recommended. From the methods description is unclear whether the feature selection/ranking has been correctly performed on a separate portion of the data to avoid unwanted selection bias effects. Similarly, it is unclear whether the cross validation has been performed at training session level or at players level. In the first case, the analysis may suffer from the data leakage effect, the overfitting effect occurring whenever both training and test portion include data belonging to the same subject (Suggestion) The paper would greatly benefit from asking a professional trainer to comment the obtained results and top-features.Minor:
Bland-Altman analysis should be described in the main text CR-10 Borg scale should be described in the main text A Table in the Main Text summarising the main results would help the reader Top features should be more extensively described and interpreted What about training a model with only the top-k features? Would it improve the performances? XGBoost may be another algorithm to compare to writings like 0.01+/-1.20 or 0.002+/-1.57 where SD is larger than mean are unconventional to say the least for positive quantities Line 255 "their intensity" is unclearTypos:
"allow to" is uncorrect lines 214-217 (Discussion...highlighted) come from the MDPI template and should be dropped Line 256 "Accordling"
Author Response
Point 1. The paper has merit, but there are issues that need to be discussed before the paper reaches publication quality level.
Authors’ response: We would like to thank reviewer 2 for the useful comments provided.. We will provide detailed answers below. All the changes are highlighted in red throughout the paper.
Point 2. Major: The paper relies on a quite small dataset: although they use 1674 training session, they belong to only 22 players. I wonder whether the analysis (and the drawn conclusions) may suffer from this small number. The experimental design only involves a 3-fold cv + 90/10 stratified split; this design may be not sufficient to warrant reproducibility - following the US-FDA guidelines exploited in the MAQC/SEQC studies, a 10x5-CV + external validation Data Analysis Plan would be recommended. From the methods description is unclear whether the feature selection/ranking has been correctly performed on a separate portion of the data to avoid unwanted selection bias effects. Similarly, it is unclear whether the cross validation has been performed at training session level or at players level. In the first case, the analysis may suffer from the data leakage effect, the overfitting effect occurring whenever both training and test portion include data belonging to the same subject (Suggestion) The paper would greatly benefit from asking a professional trainer to comment the obtained results and top-features.
Authors’ response: This kind of data are extremely difficult to obtain from the teams. Hence, it is quite difficult to have a larger dataset. We have hence added this point at the limitation section inserted at the end of the discussion session. The feature selection process was performed on 20% of the dataset, while the cross validation part was performed on the remaining 80%. The cross validation process was stratified by player ID to avoid overfitted results. We have added this information in the method section. Three of the five authors are sport scientist and two of them are professional trainers who work with Italian elite soccer clubs and who helped to comment the obtained results and top-features throughout the text.
Point 3. Minor: Bland-Altman analysis should be described in the main text. CR-10 Borg scale should be described in the main text. A Table in the Main Text summarising the main results would help the reader. Top features should be more extensively described and interpreted. What about training a model with only the top-k features? Would it improve the performances? XGBoost may be another algorithm to compare to writings like 0.01+/-1.20 or 0.002+/-1.57 where SD is larger than mean are unconventional to say the least for positive quantities. Line 255 "their intensity" is unclear
Authors’ response: Thank you for your useful suggestion. We have added some details throughout the text in order to provide all the information required. In particular, the part of the text where Bland-Altman analysis and results were already described was a little bit improved in order to be sure that the reader have all the information to understand it. The cr10 Borg scale where better described in the “Method” section. We decided not to provide tables describing all the results (that could be a lot) in the main text in order to make the text fluent for the readers. We made sure that all the results needed to understand the paper are well described into the text. All additionally information are provide in supplementary material. Finally, we tested several other ML approach during analysis but we decided to provide the results of the best ones not to provide too many information throughout the text that could made the text difficult to read.
Point 4. Typos: "allow to" is uncorrect lines 214-217 (Discussion...highlighted) come from the MDPI template and should be dropped Line 256 "Accordling"
Authors’ response: During the revision process, we deleted the sentence at line 214-217. We also correct several typo find throughout the text.
Reviewer 3 Report
This study is well-thought and carried out. The authors should be commended for the standard and number of players that were assessed as well as the quality of the study.
There are a number of things I believe they should adjust to improve the manuscript:
- pg 2
- The "why?" of this study really needs emphasising particularly in the introduction.
More about the sport specific background of internal and external load. How this can be predicted in sport and specific in soccer
For example, some of the references
Williams, G. Trewartha, M. J. Cross, S. P. T. Kemp, and K. A. Stokes, “Monitoring What Matters: A Systematic Process for Selecting Training-Load Measures,” Int. J. Sports Physiol. Perform., vol. 12, no. Suppl 2, pp. S2-101-S2-106, 2017. Robertson, J. D. Bartlett, and P. B. Gastin, “Red, Amber, or Green? Athlete Monitoring in Team Sport: The Need for Decision-Support Systems,” Int. J. Sports Physiol. Perform., vol. 12, no. Suppl 2, pp. S2-73-S2-79, 2017. Halson, L. M. Burke, G. Balagué, and D. Farrow, “An Integrated , Multifactorial Approach to Periodization for Optimal Performance in Individual and Team Sports,” pp. 538–561, 2018.
Authors should discuss the possibilities of using their solution for the automatic generation of
sport trainings in football. Authors should compare it with recent advances in single sports:
For example, some of the references:
Novatchkov and A. Baca, “Artificial Intelligence in Sports on the Example of Weight Training Rauter, “New approach for planning the mountain bike training with virtual coach,” Trends Sport Sci., vol. 2, no. 25, pp. 69–74, 2018. Fister Jr., S. Rauter, K. L. Fister, D. Fister, and I. Fister, “Planning Fitness Training Sessions Using the Bat Algorithm,” in Information technologies - application and theory, 15th conference ITAT 2015, 2015, pp. 121–126.
- pg 2 line 55 -« better explanation of the aim of the study
This is more a comment than anything - - can you think of different way to present the data?
-pg 7: Authors should mention also all limitations of this study.
- pg 7: Discussion:
Please give a more information about the practical benefits of this study.
- pg8 line 281
The conclusion is written too generally.
Author Response
Point 1. This study is well-thought and carried out. The authors should be commended for the standard and number of players that were assessed as well as the quality of the study.
Authors’ response: We would like to thank reviewer 3 for the useful comments provided. We will provide detailed answers below. All the changes are highlighted in red throughout the paper.
Point 2. pag 2: The "why?" of this study really needs emphasising particularly in the introduction. More about the sport specific background of internal and external load. How this can be predicted in sport and specific in soccer? For example, some of the references:
Williams, G. Trewartha, M. J. Cross, S. P. T. Kemp, and K. A. Stokes, “Monitoring What Matters: A Systematic Process for Selecting Training-Load Measures,” Int. J. Sports Physiol. Perform., vol. 12, no. Suppl 2, pp. S2-101-S2-106, 2017. Robertson, J. D. Bartlett, and P. B. Gastin, “Red, Amber, or Green? Athlete Monitoring in Team Sport: The Need for Decision-Support Systems,” Int. J. Sports Physiol. Perform., vol. 12, no. Suppl 2, pp. S2-73-S2-79, 2017. Halson, L. M. Burke, G. Balagué, and D. Farrow, “An Integrated , Multifactorial Approach to Periodization for Optimal Performance in Individual and Team Sports,” pp. 538–561, 2018.
Authors should discuss the possibilities of using their solution for the automatic generation of
sport trainings in football. Authors should compare it with recent advances in single sports:
For example, some of the references:
Novatchkov and A. Baca, “Artificial Intelligence in Sports on the Example of Weight Training Rauter, “New approach for planning the mountain bike training with virtual coach,” Trends Sport Sci., vol. 2, no. 25, pp. 69–74, 2018. Fister Jr., S. Rauter, K. L. Fister, D. Fister, and I. Fister, “Planning Fitness Training Sessions Using the Bat Algorithm,” in Information technologies - application and theory, 15th conference ITAT 2015, 2015, pp. 121–126.
Authors’ response: Thanks for the useful comment. We added these references throughout the paper improving the background of internal and external load analysis and the discussion section with more practical application in accordance with reviewer’s comments.
Point 3. pag 2 line 55. Better explanation of the aim of the study. This is more a comment than anything - - can you think of different way to present the data?
Authors’ response: We have had better explain the aim of this work at the end of “Introduction” section.
Point 4. pag 7: Authors should mention also all limitations of this study.
Authors’ response: We have added the limitations section at the end of the discussion section.
Point 5. pag 7: Discussion: Please have given a more information about the practical benefits of this study.
Authors’ response: Thanks for the suggestion. We have added some practical application and benefits throughout the “Discussion” and “Conclusion” sections.
Point 5. pag8 line 281: The conclusion is written too generally.
Authors’ response: We have modified the “Conclusion” section adding some detailed conclusion about our work.
Round 2
Reviewer 1 Report
NO COMMENTS
Reviewer 2 Report
All the raised issues have been reasonably met.